# Sourcing and Propagation of *Pontechium maculatum* for Horticulture and Species Restoration

**DOI:** 10.3390/biology9100317

**Published:** 2020-09-30

**Authors:** Barbara Nowak, Ewa Sitek, Joanna Augustynowicz

**Affiliations:** Department of Botany, Physiology and Plant Protection, Faculty of Biotechnology and Horticulture, University of Agriculture in Krakow, al. 29 Listopada 54, 31-425 Krakow, Poland; ewa.sitek@urk.edu.pl (E.S.); j.augustynowicz@urk.edu.pl (J.A.)

**Keywords:** *Echium russicum*, generative propagation, in vitro, medicinal plants, *Pontechium maculatum*, Russian bugloss, seed dormancy, species rehabilitation

## Abstract

**Simple Summary:**

Russian bugloss, a species of ornamental, apicultural and medicinal value, is threatened in some Central European countries. To restrict its overexploitation from nature the alternative method of propagation using tissue culture was elaborated here. Additionally, the generative reproductive ability was compared for two groups of plants obtained from seeds (received from Germany and representing the Polish vanishing population) and those of in vitro origin. It was proved that the German seed-origin plants had the greatest propagation efficacy and developed the greatest number of seeds. The ability of seeds to germinate was similar for all plants; however, seeds were in a state of dormancy, which can be broken using plant growth regulators. It was also proved that cultivation in vitro can induce some variability among received plants, which makes this way of Russian bugloss propagation a potential breeding tool.

**Abstract:**

*Pontechium maculatum*, a species of ornamental, apicultural, health and medicinal value, is threatened in some Central European countries including Poland. Its propagation using seeds or in vitro techniques is needed for multiple applications including conservation. Generative propagation efficacy of *P. maculatum* plants representing different genetic resources (received from botanical gardens in Germany and in Poland) propagated from seeds or in tissue culture was assessed. Moreover, an efficient technique of propagation of *P. maculatum* using in vitro shoot culture from seedlings was elaborated for the first time. The highest propagation efficacy was noted for German plants of seed origin. The ability of seeds to germinate was similar for all plants; however, seeds were in a state of dormancy, which was broken by GA_3_. After two years of storage, the seeds still retained the ability to germinate though seeds from propagated in vitro plants germinated more poorly than those from seed-originated plants. The ploidy assessment showed that some in vitro-origin plants had altered DNA content. The results indicate that efficacy of generative propagation of *P. maculatum* is resource dependent. Furthermore, results suggest that cultivation in vitro influenced some generative features of examined species, which makes this way of *P. maculatum* propagation a valuable source of genetic variation and a potential breeding tool.

## 1. Introduction

Natural resources of cultivated plant species are declining mostly because of their excessive exploitation. As a result, a lot of useful species and crop relatives have disappeared in the wild or their populations are endangered [1]. An alternative plant source for both applied purposes and species conservation are of great need. However, the choice of the propagule source and the method of propagation is a challenge. In the case of rare species, particularly of limited reproductive abilities or declining populations, efficient methods of propagation are of great value. In horticulture, the priority are the methods allowing to obtain great number of plants, possibly uniform, and in a relatively short period of time. The micropropagation cloning technique is such a method [2].

In turn, in case of efforts aimed at restoration of plant populations, the first concern is to ensure the greatest possible genetic variability, from which arises serious dilemmas. A crucial one is the origin of propagules. It is commonly accepted that the greatest benefit comes from using seeds from the mother population. Such strategy allows keeping an existing and often unique gene pool [3]. Using local material guarantees keeping beneficial traits of local ecotype, best suited for local conditions. On the other hand such, declining, especially isolated, populations are characterized by poor gene pool due to genetic erosion and inbreeding depression, which limits their reproduction and adaptation abilities [4]. Strategy of mixed seed sources can support, in such situations, successful species rehabilitation in the wild, and preserve them also as a gene source for the future [5,6].

Russian bugloss (*Pontechium maculatum* (L.) U.-R. Böhle and H.H. Hilger) was previously known as *Echium russicum* J.F.Gmel with chromosome number 2*n* = 24, which is treated as diploid, triploid or tetraploid [7,8]. Divergances in the assessment of plant ploidy indicate how knowledge about this species is limited, additionally emphasizing the need to learn about its propagation biology. It is a biennial or perennial plant from the Boraginaceae family. Classified as hemicryptohyte, in the first year it forms a rosette of lanceolate leaves. In the second year it develops an inflorescence which is 10–40 cm long and consists of numerous helikoid cymeses. The flowering period typically lasts 2–3 weeks and falls in either May or June. The fruit is a schizocarp that splits into four one-seeded mericarps. In the successive vegetation seasons at the base of the parental rosette, some daughter rosettes emerge, which can be used for vegetative propagation [9].

Because of a considerable amount of nectar being produced, Russian bugloss is a plant of great apicultural value. The nectar is secreted by four nectary glands located at the base of the ovary [10]. As other plants from Boraginaceae family, Russian bugloss contains valuable compounds which make it a natural source of antioxidants [11]. It contains a considerable amount of rosmarinic acid and shikonine—well-known compounds with medicinal properties [12,13]. The seeds with high content of omega-3 and omega-6 fatty acids make it a valuable plant for cultivation and production of useful fatty acids [14]. It has been used as an anti-inflammatory medicine that accelerates wound healing and is applied against snakebites [15]. 

The ornamental value of this plant, resulting from its beautiful and original inflorescences, makes it a valuable horticultural plant known as Redflowered viper’s bugloss. It is recommended by the producers in Europe and the USA as a species suitable for dry, sunlit places [16].

The Russian bugloss is a Pontic-Pannonian sub-element, which spans over Central and Eastern Europe, the Caucasus and Anatolia [17]. It occurs in many European countries, but in some regions it is an endangered species [9,16,18,19]. It is also included in Annexes II and IV of the Habitat Directive 92/43/EU (as amended by the Directive 2013/17/EU) which covers European species requiring strict protection and special areas of conservation. 

The north-western geographical range limit of *P. maculatum* crosses through Poland [17]. The Polish range of the species is located within the Lublin Upland and the Volhynian Upland. Monitoring conducted in 2013 revealed only three localities out of all known historical ones and the presence of only seven wild growing specimens. *P. maculatum* in Poland is a critically endangered (CR) species under legal protection requiring active conservation [9].

To conserve and to propagate wild crop relatives, some key information is needed, particulary that related to reproductive biology, reproductive output and variability among populations [20,21,22], which have not been investigated for *Pontechium maculatum* yet. Another approach to propagation and conservation are micropropagation techniques. The plants derived from tissue culture can be used in horticulture, the medicinal/pharmaceutical industry and as living plant collections in botanical gardens instead of plants collected from wild populations [23].

In Boraginaceae plants, tissue cultures are mainly used as a source for the production of valuable secondary metabolites. Numerous papers have described cultivation of callus and suspension, from which compounds were obtained for industrial purposes [24,25]. Secondary metabolites of medicinal properties were obtained from cotyledon-originated callus of *Echium amoenum* Fisch. and C.A. Mey., species from the Iran area [26]. Callus induced on explants excised from seedlings was a good source of shikonin [27], whereas regeneration of adventitious buds was described only for two endemic species, *E. amoenum* and *E. orientale* L. [28,29], where the differentiation took the form of indirect organogenesis which involved the callus phase. 

Propagation of plants in tissue culture is associated with the risk of emergence of somaclonal variation resulting from conditions of cultivation. Probability of such variation is difficult to estimate and it depends on many factors, such as the type of initial explants and ways of regeneration [30]. The least variation is expected in cultures initiated from seeds sown on a medium, from which shoot cultures are derived with omission of the callus. Such a way of in vitro propagation for *Pontechium* and *Echium* genus has not been described yet.

The aim of the current work was (1) description of biology of generative propagation of *P. maculatum*, with particular emphasis on seed production and their ability to germinate at the moment of harvest and after storage, for three groups of plants of different origin including in vitro-origin ones, and (2) to elaborate on an effective method of propagation from seeds using in vitro shoot culture. Two hypotheses were verified. (1) The ability for generative propagation of *P. maculatum* representing a Polish declining population differ from such abilities of the other resources of this species. (2) Propagation in vitro does not cause substantial changes limiting generative propagation.

## 2. Materials and Methods

### 2.1. Plant Material

In this study, plants obtained from seeds received from the Botanical Garden of Maria Curie-Sklodowska University in Lublin and plants from seeds received from Germany Universität Bayreuth Ökolog-Botanischer Garten were used. Seeds from Germany were used as a source material for shoot cultures in vitro. Seeds from Polish botanical garden represent Polish natural resources from no longer existing populations in Czumów (further in the manuscript described as Polish origin). Seeds obtained from German botanical garden were of unknown origin (described as German origin). 

### 2.2. Ex Vitro Plants from Seeds

Seeds of both origins were sown on 2 January 2012 into boxes (25 × 35 × 8 cm) with peat substrate and sand mixture in ratio 3:1 (Figure 1a). The germination was carried out for four weeks in Sanyo vegetative chambers, under 16/8 h day/night photoperiod and photon flux density of 60 μmol m^−2^ s^−1^, at 24/20 °C, with humidity 60%. One month later, the plants were replanted in pots and were cultivated in open field conditions in the collection of the Department of Botany, Physiology and Plant Protection of the University of Agriculture in Krakow (Kraków Poland) for one season.

### 2.3. In Vitro Propagation

#### 2.3.1. Shoot Culture

German-origin seeds were disinfected for 30 s in 70% ethanol solution and then in HgCl_2_ for 3.5 min. After rinsing three times in sterile water they were sown onto a Murashige and Skoog (MS) medium [31] with the addition of 20 g L^−1^ of sucrose (Chempur, Piekary Śląskie, Poland) and solidified with 8 g of agar (Agar-Agar S-0030, BTL, Łódź, Poland). After germination, the seedlings were cut and the excised shoots were put onto the MS medium supplemented with 1.0 mg L^−1^ of thiamine, 0.5 mg L^−1^ of nicotinic acid, 100 mg L^−1^ of myo-inositol, 2.0 mg L^−1^ of glycin, 30 g L^−1^ of sucrose, solidified with agar, pH 5.8 (basal medium); 6-benzyl-aminopurine (BAP) combined with IBA (indole-3-butiric acid) or NAA (α-naphthalene-acetic-acid) (Table 1) were used as plant growth regulators at the stage of multiplication (multiplication media). In the preliminary rooting experiment, there was no spontaneous root induction noted on the MS medium without growth regulators. Therefore, the effect of auxins, 0.1 mg L^−1^ IAA (indole-3-acetic acid), IBA or NAA at the rooting stage was investigated (Table 2). All chemicals if not mentioned otherwise were manufactured by Sigma-Aldrich (Saint-Louis, MO, USA).

Rosette explants, four-weeks old and approximately 1.0 cm high (for multiplication and rooting) with 4–5 leaves, were planted in 100 mL Erlenmeyer flasks that were kept in phytotrone with 16/8 day/night and photon flux density of 70 μmol m^−2^ s^−1^ and temperature of 24 ± 2 °C.

Efficacy of multiplication (multiplication rate expressed as number of newly developed shoots) and rooting (percentage of rooted shoots, length and number of roots) and quality of plants (height, number of leaves, vitrification) were assessed at the end of each four-week passage. In each combination there were five flasks with six explants, and the experiment was repeated three times. Samples from 20 shoots were collected for ploidy assessment.

After multiplication and rooting, the plants were replanted in pots and after acclimatization they were cultivated in open field conditions for one season. The acclimatization was carried out for four weeks in Sanyo vegetative chambers, under 16/8 h day/night photoperiod and photon flux density of 60 μmol m^−2^ s^−1^, at 22/20 °C, with humidity gradually decreasing from 90% to 55%, and then in a non-heated greenhouse.

#### 2.3.2. Indirect Organogenesis

Fragments of leaves 1.0 cm long excised from shoots cultivated on a basal medium without plant growth regulators were placed on a basal medium supplemented with BAP, IBA, NAA or IAA and GA_3_ in various combination (Table 3) and cultivated for five weeks in photoperiod conditions and photon flux density of 70 μmol m^−2^ s^−1^ or in continuous darkness. There were six explants placed with the abaxial side in contact with the medium in one 100 mL Erlenmeyer flask, with five flasks for each combination, and the experiment was repeated three times. At the end of the subculture, the percentage of callusing and regenerating explants was estimated together with the number of regenerating adventitious buds/shoots per explant. Twenty randomly chosen shoots were isolated and cultivated on a basal medium for ploidy assesment.

#### 2.3.3. Ploidy Assessment 

Plant ploidy analysis based on flow cytometry was performed in the Cytogenetics Laboratory of the Sugar Beet Breeding Station in Kutno (Poland). The ploidy level was determined for samples of young leaves collected separately from each plant (those from proliferated shoot culture and shoots regenerated on callus) and prepared according to the Galbraith method [32] modified by Thiem and Śliwińska [33]. Chopped plant material in 2 mL of lysis buffer with the addition of fluorochrome dye DAPI was filtered and analyzed using a PAII (Partec, Münster, Germany) cytometer. In each sample 720–1100 nuclei were assessed. The seed-origin plant from Germany served as a reference.

### 2.4. Comparison of Generative Potential of Plants in Open Field Study

After overwintering, during the following spring the seed-origin plants from Poland and from Germany and plants from in vitro shoot cultures were planted in groups of 15–19 in comparable habitat conditions at the collection of the university.

Over three successive vegetative years, the number of vegetative rosettes, the number of inflorescences (IP) and their length, and the number of cymeses in inflorescence (CI) were assessed. The number of flowers in cymeses (FIC) and the number of seeds set per fruit (SFr) were estimated for 30 randomly chosen cymeses or fruits from the middle section of inflorescences in each group. The data was used for the calculation of the actual production of seeds (AP), in particular plant groups, expressed as the number of seeds set per one flowering plant; one specimen was treated as repetition. Potential ability for seed production (PP) was estimated assuming that in each flower one fruit splits into four one-seed mericarps. The efficacy of seed setting was estimated as a ratio of actual production (AP) to a potential one (PP).

In the blooming period, pollen viability was assessed by the staining method [34]. In each examined group of plants, three mixtures of pollen taken from five different plants were assessed. In each preparation sample, microscopic analysis was conducted for 250–300 pollen grains; evaluation was done in three consecutive vegetative seasons.

The plants set fruit in open pollination conditions. Seeds were collected from all harvested fruit and immediately after the harvest their ability to germinate was assessed in the blotter test. Part of the seeds was soaked in 1.8 mM gibberellin (GA_3_) solution for 24 h before placing them on tissue paper. There were six Petri dishes in combination (+GA_3_, control), with 15 seeds each. Dishes were kept at a temperature of 24 ± 2 °C for four weeks under photoperiod conditions. 

The remaining seeds collected from plants were also stored at 20 °C and their ability to germinate was assessed after one and two year periods.

The observations carried out for one-year-old seeds allowed for the investigation of dynamics of their germination in all investigated groups of plants.

### 2.5. Statistical Analysis

The results were evaluated with ANOVA and Tukey’s test at *p* ≤ 0.05, treating one Petri dish, one flask, one plant or pollen mixture from five plants as a repetition. The percentage of flowering specimens was compared using multiway contingency tables and chi-square test. Statistical analyses were done using Statistica 13 (StatSoft Inc., Tulsa, OK, USA).

## 3. Results

### 3.1. In Vitro Propagation

#### 3.1.1. Shoot Culture

The multiplication rate of *P. maculatum* observed for all media was over four and was the highest on the medium without growth regulators (Figure 1b). The quality of the obtained plants was comparable. Worth mentioning is the high percentage of plants with vitrification, even on a medium with no growth regulators added (Table 1). 

The rooting ability of shoots was comparable regardless of the auxin being used (ca. 53–63%). Rooted plants had similar length of root system as well as shoot height (Table 2). The plants on media supplemented with NAA or IBA maintained multiplication ability, which was not noted for plants on IAA medium. Out of 20 plantlets selected for ploidy assessment, 15 specimens that survived acclimatization and had the same DNA content as seed-origin plants were grown in pots under open field conditions. 

#### 3.1.2. Indirect Organogenesis

Callus tissue developed as early as within five to seven days of cultivation both on distal and proximal cut edges and numerous adventitious shoots started to differentiate, develop and grow. Continuous darkness positively affected both the efficiency of callus induction and the process of differentiation of adventitious buds, which took the form of indirect organogenesis in all cases. The addition of plant regulators was beneficial for regeneration because it increased the percentage of regenerating explants and the number of developing buds compared to the medium without growth regulators. The highest regeneration (over 80%) was noted in media supplemented with either 0.5 or 0.2 mg L^−1^ of BAP. The addition of gibberellic acid (GA_3_) improved regeneration abilities and it was the only medium on which the adventitious buds developed in photoperiod conditions (Table 3, Figure 1c); GA_3_ also prevented explants cultivated in the photoperiod from browning. Explants cultivated on the other media in photoperiod conditions turned brown within two weeks of culture.

### 3.2. Ploidy Assessment

Results of flow cytometric analysis presented on DNA histograms (Figure 2) showed a distribution of relative DNA content with dominant peaks corresponding to the 2C level in the G1 phase of the cell cycle of the seed-origin plant from Germany (control plant, Figure 2a); in vitro-origin after indirect organogenesis is shown in Figure 2b and shoot culture in Figure 2c. The DNA content analyzed for all plants after indirect organogenesis indicated that they did not differ from seed-origin plants, while among plants after shoot culture two specimens turned out to have DNA content two times higher. These two plants were not assessed in the collection because they did not survive during first winter in the field conditions

### 3.3. Comparison of Generative Potential of Plants in Open Field Study

All of the *P. maculatum* plants were perennials (Figure 1d)—only one specimen from the German group died in the second year. The number of descendant rosettes developed from buds on the roots of mother plants was comparable in all groups.

In the first year only the plants originating from Poland developed vegetative rosettes. However, in the second year in vitro plants from Germany began to grow more intensively. The number of blooming specimens varied in the consecutive seasons due to different weather conditions. However, on average the lowest number of blooming specimens was among Polish-origin plants. They also developed the fewest inflorescences (Table 4).

Potential production of seeds (PP) varied between seasons, but on average the highest was for German plants of seed origin. In the other two groups, this potential was lower, which was the result of different factors: In in vitro-origin plants it was caused by a smaller number of flowers in cymeses, and in Polish-origin plants due to the smaller number of cymeses in inflorescences.

It was also noted that flowers on plants representing the Polish population had less vivid color of corolla (Figure 1e,f).

Additionally, the observed efficacy of seed setting for plant from in vitro was 18.8% and from Poland 26.8%, which was lower than for seed-origin plants from Germany. This means that worse features crucial for potential and actual ability for generative propagation were noticed for Polish-origin plants and for in vitro-origin plants (Table 4, Table 5 and Table 6). Generally, the efficacy of seed setting noted for plants in all groups was low.

#### 3.3.1. Pollen Viability

Pollen viability was within the 45–70% range. The results were diversified depending on the origin of the plants (Figure 1g,h). Pollen viability also varied in each consecutive season (Table 7).

#### 3.3.2. Germination Ability of Seeds

The ability of seeds collected from plants of different origin to germinate in the year of harvest was comparable and was in the 56–68% range. This ability was higher for seeds that were stored for one year and was within 64–78%, being comparable for each type of seed. However, seeds collected from in vitro plants after storage for two years had a lower germination rate than seeds collected from seed-origin plants regardless of their origin. Because the application of gibberellin did not improve the germination ability in this case, it can be assumed that these seeds began to lose their vitality, quite different than seeds from plants of seed origin.

The beneficial influence of gibberellin on germination ability was noticed for seeds assessed in the year of harvest. It increased the percentage of germinating seeds (Figure 1i and Figure 3; Table 8), without significant influence on the dynamics of germination: Most seeds germinated by eighth day, with the exception of German seeds without gibberellin treatment and Polish-origin plants treated with GA_3_. In those groups germination took place up to 28 days after sowing. 

## 4. Discussion

Effective propagation is a key factor for profitability of horticultural production and survival of species in their habitat. Propagation using seeds in commercial production could be most convenient, providing seed production is abundant, they are viable and easily germinate even after storage period. Additionally, in wild populations, effective generative propagation promotes adaptability of species. In turn, micropropagation is considered as a cloning technique resulting in numerous uniform specimens, which is a great advantage in horticulture.

Presented results for the first time describe selected aspects of generative propagation of *P. maculatum*, which is an endangered species in numerous countries. Efficiency of generative propagation was assessed between two European genetic resources of this species originating from Poland and Germany. The method of in vitro propagation from seeds and leaf explants was developed and obtained plants were evaluated for generative propagation changes generated in tissue culture.

As presented in this paper, a comprehensive procedure of propagation in shoot cultures from seeds is the first in vitro propagation method of *Pontechium* and *Echium* species with omission of the callus phase. The obtained multiplication coefficient makes it an efficient method. The best multiplication rate for *P. maculatum* was noted on the medium without growth regulators (8.9 shoots) or on the medium with 0.1 mg L^−1^ NAA designed for rooting (16 shoots). It seems that the species does not need cytokinins added to the medium to multiply. It can be the result of a high level of endogenic plant growth regulators and can be the result of natural abilities to develop vegetative rosettes. Cytokinins are produced mainly in roots [35] and this could be a reason that rooted plants multiplied better than cultivated shoots. The other species of the Boraginaceae family with a great multiplication rate have already been reported. The number of new developed shoots reached 7.1 for *Heliotropium indicum* Linn., 17.7 for *Mertensia maritima* L. and 20.1 for *Arnebia benthamii* Wall [36,37,38]; however, they multiplied on a media supplemented with auxins and cytokinins. It is debatable how long the ability of effective cloning without plant growth regulators would be sustained in continuous culture. Easy acclimatization together with great regenerative abilities of the species in the experimental conditions presented here makes tissue culture cloning an effective propagation method for examined species. 

Unlike shoot multiplication, the addition of plant growth regulators was beneficial for callus induction and organogenesis on leaf explants (Table 3). The callusing and regeneration were rapid and profuse on explants cultivated in darkness, whereas photoperiod conditions suppressed differentiation. This adverse effect of light is thought provoking, although darkness-promoting somatic embryogenesis [39], bulblet regeneration [40] and rooting [41] have been previously observed. It was also proved that early exposure to light can prevent callus formation and decrease adventitious shoot regeneration [42]. Light is the most important factor which regulates plant growth and development in vitro because it is involved in photomorphogenic responses [43]. For this reason, the mechanisms causing the negative influence of light on cultivated tissues can be complex and still poorly understood. We observed that leaf explants cultivated in photoperiod conditions were browning, which is a physiological disorder observed in many plant species during in vitro propagation [44]. It was reported that some explants release phenolic substances as secondary metabolites from cut surfaces. They can further oxidize and form quinone derivatives that can be toxic to the explants [45]. Moreover, light treatment during cultivation can be a reason of photooxidative stress responsible for the increased level of reactive oxygen species (ROS), which contribute to phenol oxidation [42]. The regeneration abilities of *P. maculatum* explants cultivated in photoperiod were improved when GA_3_ was added to the medium. It can be hypothesized that GA_3_ treatment activated some ROS scavengers to reduce oxidative stress or induced the increased accumulation of proteins and amino acids (e.g., proline), which can counteract stress symptoms. Recently published data demonstrated that GA controls certain physiological processes in response to stress [46] and the positive effect of gibberelline treatment on the antioxidant complex was described for *Picea asperata* Mast. [47]. 

The assessed ploidy level of plants resulting from shoot cultures and microshoots developing from adventitious buds on callus indicates that shoot cultivation induced variation, whereas it was not observed in the latter. Somaclonal variation occurring uncontrollably in tissue culture is more frequent usually in regenerants obtained from callus [48]. Our results show that *P. maculatum* is prone to variability caused by conditions of in vitro cultivation, even if the callus stage is omitted. Somaclonal variants were already reported in shoot cultures of different species [49,50,51]. This lack of genetic stability can be recognized as disadvantageous for conservation purposes. On the other hand, it is believed that it could be beneficial in the restitution of a small population [52,53]. Nevertheless, new genetic variation makes this propagation method a valuable source of new qualities potentially useful in horticulture. 

In all assessed groups of *P. maculatum*, the plants were perennials which developed descendant rosettes and bloomed every year. Pollen viability ranging between 50.7% and 67.4% was lower than observed for *E. vulgare* L. [54]. Notwithstanding, observed pollen viability was not the factor determining the efficacy of fruit setting (Table 5; Table 7). Depending on the plant group tested, plants of *P. maculatum* developed on average 0.8–1.1 seeds from one flower in conditions of open-field pollination. That means that approximately one of four fruit mericarps contained a seed similarly as it was described for *E. vulgare* [55]. However, the observed germination ability was higher than that reported for *E. vulgare* [56,57]. The seeds of *P. maculatum* germinated rapidly and synchronously independently of GA_3_ treatment and their germination ability was sustained until the third year after harvest. The influence of gibberellin as a stimulating agent of germination of *P. maculatum* was limited to seeds sown in the year of harvest when it accelerated germination and in most cases enlarged the percentage of germinating seeds. This indicated the transition into post-harvest dormancy, which most likely ended after one year. GA_3_ applied one and two years after the harvest did not increase the germination ability of seeds. Gibberellin, natural phytohormone, is known for its ability to reduce seed dormancy time and successfully stimulated seed germination of other plants [58].

Comparison of two groups of *P. maculatum* (Polish and German) showed that they differ in the features that determine the potential and actual generative propagation ability. Altered structure of the inflorescence of Polish-origin plants restricted mainly their potential generative ability. Lower actual reproductive ability within this group was additionally exacerbated by decreased efficacy of fruit setting. The efficacy of seed setting in all compared groups was within 18.9–29%, the highest in German seed-origin plants, the lowest in German in vitro-origin plants. Similar seed setting ability was described for *E. vulgare* (on average 0.91–1.7 per flower), varying for populations growing in different locations [55]. Such variation could be ascribed to both varied habitats and various, genetic-dependent, reproductive abilities.

In research presented here, *P. maculatum* plants were growing in comparable conditions, on adjacent plots. Thus, maternal effect and/or limited resources cannot be the reason of lower efficacy of seed setting by German in vitro plants and Polish-origin plants. The environment of the mother plant can affect the progeny features. Environmental maternal effects have been well documented for early developmental stages such as weight of seeds or their germination ability, but they usually vanish in later stages of the life cycle and in the next generation [59]. Nevertheless, maternal effects lasting for an entire life cycle [60] and even into the next generation [61] were also reported. However, it could be assumed that the selection of the same size seedlings of Russian bugloss in the presented research most likely eliminated diversity due to the maternal effect. Another cause of different reproductive abilities of compared *P. maculatum* types could be interpopulation genetic variation. This would indicate an unfavorable change within reproductive traits of Polish-origin plants. 

For years Polish populations have been low in number. In addition, they are island populations, some 150–200 km away from the nearest located in Ukraine [62]. This limits interpopulation gene flow due to restricted seed dispersal and pollen transfer. Strong selection pressure in the face of biotic and abiotic conditions in particular locations can cause the emergence of ecotype with a set of characteristics vital to genetic distinctiveness [63]. The unique gene arrangements of these populations decided their survival and should be preserved during different biodiversity conservation programs. However, in the presence of selection pressure of isolated populations with limited exchange of genetic material, unfavorable phenomena can occur, influencing their intrapopulation genetic variability. Then, results of inbreeding depression, genetic erosion or genetic drift are pronounced [4,64]. Survival of such populations is questionable and the results of conducted research indicate that the state of the Polish population of *P. maculatum* restricts their reproductive and survival abilities. 

Our results showed that, although in vitro plants of *P. maculatum* did not differ in number of descendant rossettes from German seed-origin plants (Figure 1d), cultivation in vitro restricted their generative propagation ability and effectiveness of setting seeds. Moreover, seeds collected from in vitro plants also lost their germination abilities earlier compared to the other groups. The differences were not of a transient nature and were sustained for three consecutive years. Changes resulting from tissue culture propagation in the case of *P. maculatum* were not favorable and were the effect of somaclonal variation occurring uncontrollably during tissue culture cultivation [48]. Because genetic fidelity assessed with the help of molecular techniques relates only to fragments of the genome, not necessarily connected with heredity, the comprehensive profile of plants obtained from micropropagation should also include ploidy assessment and, above all, quality characteristics. It is possible that, regardless of the proved genetic fidelity on the molecular level, the plants obtained from tissue culture will differ from mother plants in quality features [65]. It was also demonstrated that plants with altered genome posses the same reproductive capability as a mother plant and plants growing in natural populations [53]. The changes noticed in plants obtained from micropropagation could be permanent [66] or transient of epigenetic character [30,67]. There are also cases when some changes are permanent and others decline [68].

Propagation in tissue cultures is a way of obtaining a large number of plants in a short time from a small amount of available initial material. This makes such techniques extremely valuable in case of the propagation of plants which are rare but it is also commonly applied for cultivated crops. The wild resources of plants have genetic diversity useful for developing more productive, nutritious, ornamental or resilient crop cultivars, which can be enhanced by somaclonal variation of plant tissue culture. The results for *P. maculatum* indicate that this species can not only be efficiently propagated in vitro, but is also prone for spontaneous variation. 

In efforts aimed at augmentation of existing natural populations or creating secondary localities, it is recommended to use plants obtained from seeds from populations that have undergone protective treatment. This allows the preservation of the local ecotype which is best suited for local conditions and maintaining high biodiversity within the species. However, limited reproductive abilities of Polish-origin *P. maculatum* raises some doubts about its adaptive and survival abilities in the natural habitat. 

Recently other options have been discussed when it comes to extremely small populations with little genetic variability. Falk and Holsinger [69] suggested to use seeds obtained from 3–5 populations in reintroduction. Vergeer et al. [70] recommended mixing non-local populations in case local populations with sufficient genetic variability are not available. It was also suggested to use similar ecological habitats as a source of seeds [71]. 

However, it always should be kept in mind that outbreeding depression can occur as a consequence of interpopulation crosses that decreases offspring conditions [72,73]. It should also be taken into account that existing gene arrangements can fail in environments struggling with climatic changes [74]. All that causes that planning of restitution efforts is very challenging. It requires a balance between the value of local adaptation with the need for future adaptation potential. Beside strict local provenancing of plant material used for restoration, the strategy matching future climate (i.e., predictive provenancing) should be taken into consideration. Different approaches of seed sourcing, depending on the availability and similarity of ecotypes, also taking into account the likelihood of negative outcomes such as outcrossing depression, have been already described: Relaxed local provenancing, composite provenancing and admixture provenancing [5,6].

## 5. Conclusions

*Pontechium maculatum* plants can be effectively propagated by seeds; however, differences in this respect among available resources of this species appear. Irrespective of origin, the seeds have great germination ability which can be sustained for at least two years and after-harvest dormancy can be broken by GA_3_ treatment. 

*P. maculatum* can be efficiently propagated using tissue culture techniques both from seeds in shoot culture or using secondary leaf explants through indirect organogenesis. In vitro cultivation induced variation which can be the source of new features. In vitro-origin plants can be used for horticulture and apiculture purposes, reducing overexploitation of natural resources. 

*P. maculatum* is a rare species that requires the implementation of different active conservation programs. Presented results demonstrate that, within Polish populations of this species, deterioration of reproductive abilities occurred. Therefore, the dilemma that has to be grappled with is what source of plant material and which type of propagation for the restitution of *P. maculatum* should be used that permits both conservation of local biodiversity and sufficient reproduction capability necessary for its survival. 

## Figures and Tables

**Figure 1 biology-09-00317-f001:**
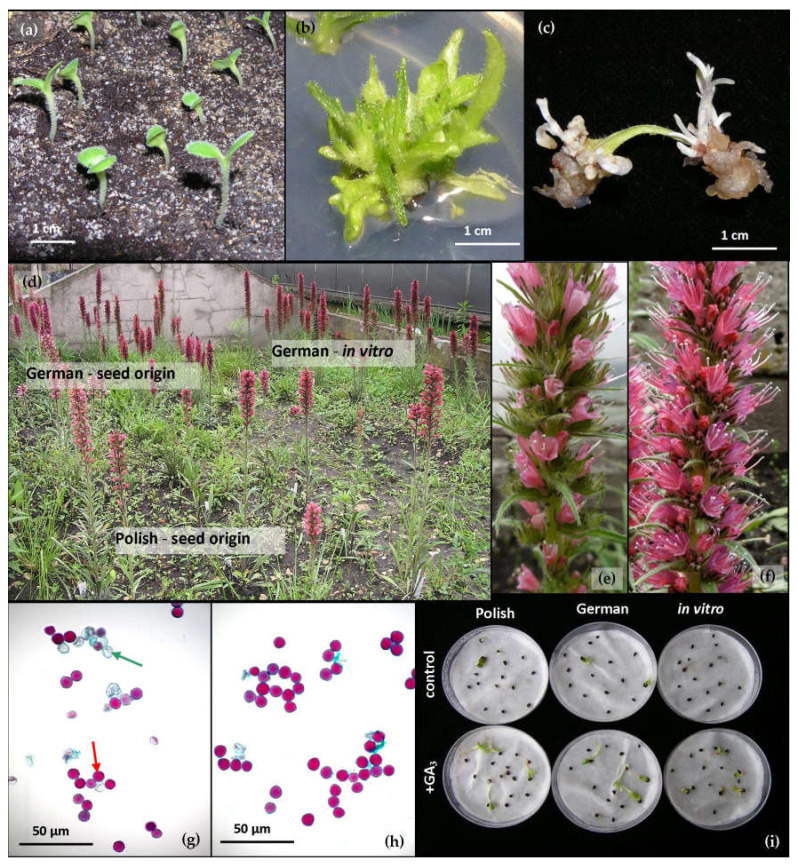
*Pontechium maculatum*: (**a**) German-origin seedlings; (**b**) shoots after three weeks of cultivation on multiplication medium Murashige and Skoog (MS) supplemented with 0.05 mg L^−1^ IBA and 0.25 mg L^−1^ BAP; (**c**) leaf explants after four weeks of cultivation on medium MS + 0.1 mg L^−1^ NAA + 0.5 mg L^−1^ BAP; (**d**) plants growing in collection; (**e**) inflorescence of Polish seed-origin plant; (**f**) inflorescence of seed-origin plant from Germany; pollen viability of Polish (**g**) and German (**h**) seed-origin plants; (**i**) germination of seeds in blotter test in the year of harvest five days after sowing. Green arrow: Dead pollen grains; red arrow: Viable pollen grains.

**Figure 2 biology-09-00317-f002:**
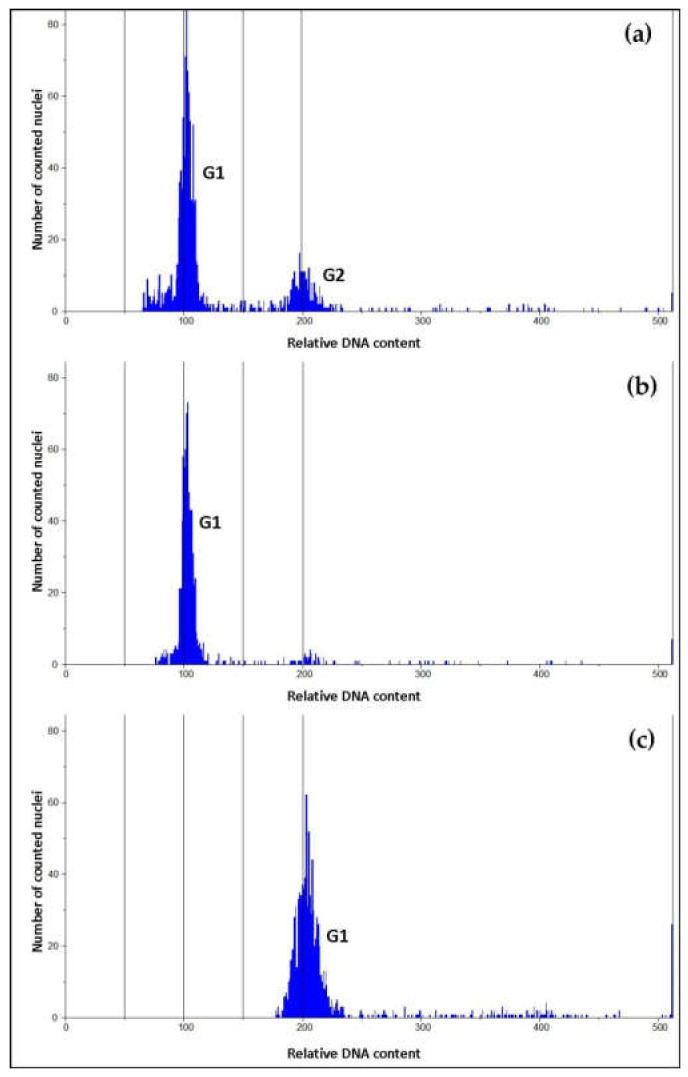
Histograms of relative DNA content in the nuclei of *Pontechium maculatum* leaf cells: (**a**) Control seed-origin plant from Germany; (**b**) plant originating from in vitro (indirect organogenesis); (**c**) one of two plants originating from shoot culture with doubled DNA content. G_1_, G_2_—phases of cell life.

**Figure 3 biology-09-00317-f003:**
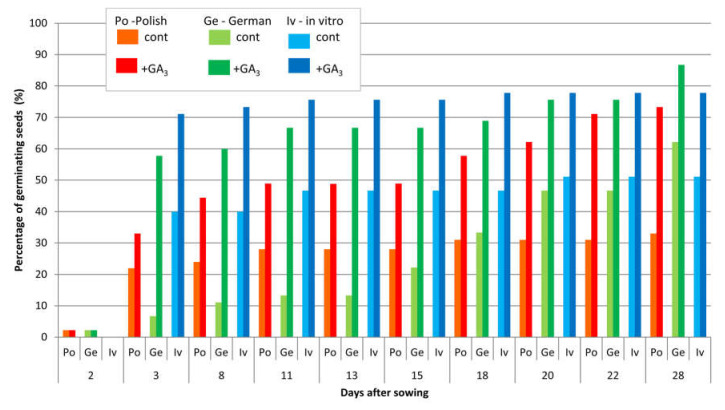
Influence of GA_3_ treatment on germination dynamics of *Pontechium maculatum* seeds of different origin in the year of harvest. (Ge)—German seed-origin plants; (Po)—Polish seed-origin plants; (Iv)—German in vitro-origin plants; (cont)—control without GA_3_ pretreatment; (+GA_3_)—seeds with gibberellin pretreatment.

**Table 1 biology-09-00317-t001:** Multiplication rate (number of shoots) and the quality of *P. maculatum* explants after four weeks of cultivation on an MS medium with plant growth regulators added.

BAP (mg L^−1^)	Auxins (mg L^−1^)		Mean Multiplication Rate	Mean Height (mm)	Mean Number of Leaves Per Shoot	Vitrification (%)
	IBA	NAA				
**-**	**-**	**-**	8.9 b *****	12.3 a	3.4 a	9.1 a
0.1	0.05	-	5.2 ab	18.7 a	6.2 a	22.8 a
0.25	-	0.05	5.6 ab	15.4 a	4.8 a	19.9 a
1.5	0.1	-	4.9 a	14.6 a	4.9 a	6.8 a
0.5	-	0.1	4.4 a	21.7 a	4.8 a	18.6 a

a, b * values within columns followed by the same letter do not differ significantly for *p* ≤ 0.5.

**Table 2 biology-09-00317-t002:** Rooting and the quality of *P. maculatum* explants after four weeks of cultivation on MS medium with auxins applied.

Auxin (mg L^−1^)	Rooting (%)	Mean Length of Root (mm)	Number of Root	Mean Height (mm)	Mean Number of Leaves Per Shoot	Vitrificatio *n* (%)	Mean Multiplication Rate
0.1 IAA	63.8 a *	29.6 a	1.9 a	29.6 a	4.7 a	3.5 a	1.0 a
0.1 IBA	59.0 a	42.4 a	4.5 a	16.6 a	5.4 a	10.7 a	6.5 a
0.1 NAA	53.8 a	22.0 a	5.7 a	14.8 a	4.5 a	10.1 a	16.0 a

a, b * values within columns followed by the same letter do not differ significantly for *p* ≤ 0.5.

**Table 3 biology-09-00317-t003:** Effect of media and light conditions on the morphogenetic response of leaf explants of *Pontechium maculatum* after five weeks of culture on media supplemented with different plant growth regulators (mg L^−1^).

Light Conditions	Medium						Mean
	MS	MS +0.1 IBA +0.5 BAP	MS +0.05 IBA +0.2 BAP	MS +0.1 NAA +0.5 BAP	MS +0.1 IAA +0.05 BAP	MS +0.1 IAA +0.05 BAP +0.5 GA_3_	
Percent of callusing explants							
Photoperiod	0.0 a * ± 0.0	0.0 a ± 0.0	0.0 a ± 0.0	0.0 a ± 0.0	0.0 a ± 0.0	0.0 a ± 0.0	0.0 A
Darkness	23.1 c ± 17.0	14.6 cd ± 4.9	16.7 bc ± 6.9	9.0 ab ± 4.8	57.5 d ± 10.5	0.0 a ±0.0	20.1 B
Mean	11.6 B	7.3 AB	8.4 AB	4.5 AB	28.8 C	0.0 A	
Percent of explants with organogenesis							
Photoperiod	0.0 a ±0.0	0.0 a ± 0.0	0.0 a ± 0.0	0.0 a ± 0.0	0.0 a ± 0.0	30.0 b ± 21.5	5.0 A
Darkness	29.9 b ± 12.1	85.4 d ± 4.9	83.3d ± 6.8	82.1 d ± 9.6	42.5 bc ± 10.5	65. 5cd ± 22.7	64.7 B
Mean	14.9 A	42.7 B	41.7 B	41.1 B	21.3 A	47.8 B	
Mean number of buds per explants (pcs)							
Photoperiod	0.0 a ± 0.0	0.0 a ± 0.0	0.0 a ± 0.0	0.0 a ± 0.0	0.0 a ± 0.0	2.4 bc ± 0.9	0.4 A
Darkness	1.0 ab ± 0.6	6.3 e ± 1.6	3.8 cd ± 1.4	4.9 de ± 1.7	3.3 cd ± 0.7	3.1 cd ± 1.3	3.7 B
Mean	0.5 A	3.2 C	1.9 BC	2.5 BC	1.7 AB	2.8 BC	

a, b, c, d, e, A, B, C * values within columns and rows followed by the same letter do not differ significantly for *p* ≤ 0.5.

**Table 4 biology-09-00317-t004:** The number of vegetative rosettes, the percentage of flowering plants, the number and length of inflorescences of *P. maculatum* plants depending on their origin.

Plant Origin	Mean Number of Vegetative Rosettes Per Plant				Flowering Specimens ** (%)				Mean Number of Inflorescences Per Flowering Plant IP				Length of Inflorescences (cm)			
	2013	2014	2015	Mean	2013	2014	2015	Mean	2013	2014	2015	Mean	2013	2014	2015	Mean
Polish—seed origin	2.2 ab *	3.7 ab	3.2 ab	3.0 A	77.8 a	44.4 a	61.1 a	61.1 A	1.6 a	3.3 ab	3.9 ab	2.8 A	12.7 a	18.2 a	18.8 a	16.1 A
German—seed origin	0.0 a	5.5 b	5.0 b	3.7 A	100 a	33.3 a	92.3 b	75.2 AB	2.9 ab	2.6 ab	7.8 ab	4.8 B	17.7 a	18.0 a	21.4 a	16.6 A
German—in vitro	0.0 a	5.3 b	3.9 ab	3.1 A	100 a	77.8 a	100 b	92.6 AB	3.8 ab	3.6 ab	6.2 ab	4.6 AB	19.7 a	14.9 a	15.1 a	19.2 A
Mean	0.9 A	4.7 B	4.2 B		90.4 B	47.6 A	80.9 B		2.6 A	3.2 A	6.1 B		16.3 A	17.4 A	19.8 A	

a, b, A, B * values within columns and rows for number of vegetative rosettes, inflorescence per flower and length of inflorescence followed by the same letter do not differ significantly for *p* ≤ 0.5. ** for 2013 *p* = 0.05248; for 2014 *p* = 0.10117; for 2015 *p* = 0.01654; means for plant origin *p* = 0.00536; means for years *p* = 0.00002.

**Table 5 biology-09-00317-t005:** The number of cymes per inflorescence and per plant, the number of flowers in cymes, the number of seeds set per one fruit of *P. maculatum* plants depending on their origin.

Plant Origin	Mean Umber of Cymes Per Inflorescence CI				Mean Number of Cymes Per Plant CP = IP × CI				Mean Number of Flowers in Cymes FlC				Number of Seeds Set per One Fruit SFr			
	2013	2014	2015	Mean	2013	2014	2015	Mean	2013	2014	2015	Mean	2013	2014	2015	Mean
Polish—seed origin	42.4 a *	23.3 a	40.6 a	37.1 A	63.4 a	74.7 a	186 ab	107.0A	9.1 b	7.0 ab	7.7 ab	8.0 B	0.9 a	1.3 ab	0.98 ab	1.0 AB
German—seed origin	63.5 b	40.9 a	47.9 a	53.9 B	172.6 ab	76.4 ab	411.9 b	252.3B	8.2 b	9.6 b	6.8 ab	8.0 B	0.8 a	1.6 b	1.20 ab	1.1 B
German—in vitro	68.3 b	44.6 a	38.6 a	50.2 AB	229. ab	130.7 ab	259.9 ab	212.1AB	4.42 a	7.1 ab	5.9 ab	5.8 A	0.6 a	0.8 a	0.88 a	0.8 A
Mean	56.4 B	35.1 A	42.9 A		143.6 A	94.7 A	295.1 B		7.3 A	7.9 A	6.8 A		0.8 A	1.1 B	1.0 AB	

* explanation: See Table 4. IP—number of inflorescences per flowering plant (Table 4).

**Table 6 biology-09-00317-t006:** Potential seed production and efficacy of seed setting of *P. maculatum* plants depending on their origin.

Plant Origin	Number of Seeds Set Per Flowering Plant AP = CP × FlC× SFr				Potential Seed Production Per Flowering Plant PP = CP × FlC × 4				Efficacy of Seed Set (%)			
	2013	2014	2015	Mean	2013	2014	2015	Mean	2013	2014	2015	Mean
Polish—seed origin	490.0a *	679.3a	1403.6ab	840.4A	2305.9a	2090.2a	5208.3a	3221.1A	21.3	32.5	27.0	26.8
German—seed origin	1174.9a	1196.1ab	3361.0b	2093.3B	5662.0a	2935.3a	13180.6b	8210.7B	20.8	40.8	25.5	29.0
German—in vitro	628.9a	705.3a	1349.4ab	921.4A	4058.4a	3712.3a	6133.6ab	4735.7AB	15.5	19.0	22.0	18.8
Mean	797.7A	817. 6A	2159.9B		4045.4A	2869.2A	8601.2B		19.2	30.7	24.8	

* explanation: See Table 4. CP—number of cymes per plant; FlC—number of flowers in cymes; SFr—number of seeds set per one fruit (Table 5); AP—actual production of seeds; PP—potential production of seeds.

**Table 7 biology-09-00317-t007:** Viability of *P. maculatum* pollen depending on plant origin in the years 2013–2015.

Plant Origin	Pollen Viability (%)			
	2013	2014	2015	Mean
Polish—seed origin	45.6 ab*	48.1 ab	58.5 bcd	50.7 A
German—seed origin	63.1 cd	51.3 ab	70.2 d	61.5 B
German—in vitro	69.6 d	44.8 a	87.8 e	67.4 C
Mean	59.4 B	48.0 A	72.1 C	

a, b, c, d, e, A, B, C * values within columns and rows followed by the same letter do not differ significantly for *p* ≤ 0.5.

**Table 8 biology-09-00317-t008:** Germination ability of *P. maculatum* seeds depending on their origin and gibberellin treatment in three consecutive years after harvest (%).

Plant Origin	Control	+GA_3_	Mean
In the year of harvest			
Polish—seed origin	41.7 a *	71.6 ab	56.6 A
German—seed origin	53.3 ab	83.3 b	68.3 A
German—in vitro	48.4 a	73.6 ab	61.0 A
Mean	47.8 B	76.2 A	
One year after harvest			
Polish—seed origin	74.0 a	81.1 a	77.8 A
German—seed origin	72.2 a	77.8 a	75.0 A
German—in vitro	58.9 a	69.7 a	64.3 A
Mean	76.2 A	68.5 A	
Two years after harvest			
Polish—seed origin	88.7 b	86.7 b	87.7 B
German—seed origin	73.3 b	100 b	86.7 B
German—in vitro	46.6 a	46.6 a	46.6 A
Mean	69.5 A	77.7 A	

a, b, A, B * values within columns and rows for one vegetation season followed by the same letter do not differ significantly for *p* ≤ 0.5.

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
