# Peer review of "Sourcing and Propagation of Pontechium maculatum for Horticulture and Species Restoration"

_biology, 2020, doi:10.3390/biology9100317_

Round 1

Reviewer 1 Report

The manuscript “Sourcing and propagation of Pontechium maculatum for horticulture and species restoration” describes an efficient propagation method in order to save its vanishing resources in nature. The results described in the paper are interesting and potentially valuable. However, some improvements and modifications are necessary before it can be accepted for publication.

Following are some suggestions for improvement:

L35-37: Citation required.

L48: Russian Bugloss (Pontechium maculatum (L.) U.-R. Böhle & H. H. Hilger) was

L84-87: Citation required.

L117: Please indicate the size of the box and the environmental conditions for seed germination.

Figure 1c: Please correct the scale bar value (The authors used 1 cm leaf segment).

Figure 1g, h: Please improve the figure quality and indicate viable and dead pollen.

L123: Murashige and Skoog (MS)

L127: Please delete “Bar = 50 µM”

L135-137: Please indicate the size and age of the explant for shoot multiplication and rooting. Also indicate the control medium for rooting.

L144: Please provide detailed methods for acclimatization.

L149: Indicate the light intensity.

L201: What about the control medium?

Table 2: Include the control.

Table 4: Flowering specimens (Include statistical analysis)

L310: multiplication coefficient? Please indicate the coefficient value in tables.

L311-315: Please compare the results with other members of Boraginaceae.

L333: GA3 is well-known to promote adventitious shoot bud induction. The promotive effect of auxin-cytokinin-GA3 combination on shoot regeneration is well documented in several plant species.

Reviewer 2 Report

Dear Authors,

the manuscript ‘Sourcing and propagation of Pontechium maculatum for horticulture and species restoration’ deals with the possibility to use in vitro culture of this threatened species for conservation activity.

The results reported in the manuscript reveal a lot of work has been done (much data and many tables). Nevertheless, the manuscript lacks of overall clarity. It is not easy to understand the main meaning of the work done. If your goal was to investigate the possibility to propagate asexually the P. maculatum, you should have micropropagated individual seedlings you obtained from the seeds. Each seedling represented a different genotype (since was derived from a sexual cross), and you had to evaluate each seedling progeny, multiplying in vitro each of them. Moreover, you should have evaluated in the field the in vivo performance of each progeny (from the single seedlings).

I suggest you re-write the manuscript, taking into account these main issues, and the other points of weakness I have indicated as comments in the attached file.

Reviewer 3 Report

Report for Biology-ID: 909512

 The authors carried out the current study entitled “Sourcing and propagation of Pontechium maculatum for horticulture and species restoration”. The study has a very interesting topic of relevance and general interest to the readers of the journal. I found the paper to be overall well prepared and I felt confident that the authors performed a careful and thorough experiment and spectral processing. But I have some minor comments.

 Abstract:

Line 10 please change curative value to….. health and medical value

Line 13 please change The manuscript presents to The current study presents

Line 24 please change German ones to German plants

 Keywords:

I recommend the author to include keywords that more specific to the study such as Pontechium maculatum

Introduction

Line 69-70… The author is advised to use fewer citations

Lines  78-83 please add a citation

Line 99 please change The aim of the presented research to ….the aim of the current work

Materials and Methods

Line 109 please change In the research plants obtained from seeds received from .. to   …In this study, plants obtained from seeds were received from

Line 117 please change seed to seeds

Line 117 please write the date as following January 2, 2012

Line 175 please change by dyeing method to…. by staining method

Discussion

Line 298 to 302 : not clear please revise this part and write it in a way to be easy to understand

Conclusion

In general, this section is very important to the reader. The author is advised to rewrite it more clear and avoid using long sentences to be easy to understand.

Line 449 please change Because p. maculatum …. To the fact that, p. maculatum is a rare species that raise the need for implementation

Line 452-456 please revise this part and fix it.

Round 2

Reviewer 2 Report

Dear Authors,

The manuscript has been improved. However, I still have the impression you refer to micropropagation in the wrong way (see comment on the attached file).

I also put some comments in order to do some additional changes.

If it is possible, I advise you to remove some references, as there are too many references, it is not a review.

Kind regards
